# Neuropsychology of Aesthetic Judgment of Ambiguous and Non-Ambiguous Artworks

**DOI:** 10.3390/bs7010013

**Published:** 2017-03-18

**Authors:** Maddalena Boccia, Sonia Barbetti, Laura Piccardi, Cecilia Guariglia, Anna Maria Giannini

**Affiliations:** 1Department of Psychology, Sapienza University of Rome, via dei Marsi 78, 00185 Rome, Italy; sonia.barbetti@uniroma1.it (S.B.); cecilia.guariglia@uniroma1.it (C.G.); annamaria.giannini@uniroma1.it (A.M.G.); 2Cognitive and Motor Rehabilitation Unit, IRCCS Fondazione Santa Lucia of Rome, 00179 Rome, Italy; 3Department of Life, Health and Environmental Sciences, L’Aquila University, 67100 L’Aquila, Italy; laura.piccardi@cc.univaq.it

**Keywords:** artists, art and brain, neuroaesthetics, brain damage, brain lesion

## Abstract

Several affective and cognitive processes have been found to be pivotal in affecting aesthetic experience of artworks and both neuropsychological as well as psychiatric symptoms have been found to affect artistic production. However, there is a paucity of studies directly investigating effects of brain lesions on aesthetic judgment. Here, we assessed the effects of unilateral brain damage on aesthetic judgment of artworks showing part/whole ambiguity. We asked 19 unilaterally brain-damaged patients (10 left and 9 right brain damaged patients, respectively LBDP and RBDP) and 20 age- and education-matched healthy individuals (controls, C) to rate 10 Arcimboldo’s ambiguous portraits (AP), 10 realistic Renaissance portraits (RP), 10 still life paintings (SL), and 10 Arcimboldo’s modified portraits where only objects/parts are detectable (AO). They were also administered a Navon task, a facial recognition test, and evaluated on visuo-perceptual and visuo-constructional abilities. Patients included in the study did not show any deficits that could affect the capability to explore and enjoy artworks. SL and RP was not affected by brain damage regardless of its laterality. On the other hand, we found that RBDP liked AP more than the C participants. Furthermore, we found a positive correlation between aesthetic judgment of AP and visuo-perceptual skills even if the single case analyses failed to find a systematic association between neuropsychological deficits and aesthetic judgment of AP. On the whole, the present data suggest that a right hemisphere lesion may affect aesthetic judgment of ambiguous artworks, even in the absence of exploration or constructional deficits.

## 1. Introduction 

Artworks may be defined as “skillful and creative expressions of human experience, in which the manner of creation is not primarily driven by any other function” [1] (p. 380), but they are also an unequivocal expression of intentionality and communication [2]. Artworks have been hypothesized to constitute a primary source of aesthetic experience and neural underpinning of aesthetic experience of artworks has been widely investigated by studies of the cognitive neuroscience of art [3]. Capitalizing on visual neuroscience studies, Chatterjee [4] developed a theoretical model of the cognitive and affective processes involved in visual aesthetic experience (VAE). According to this model, VAE is the result of a series of information-processing stages. First, all of the elementary and intermediate visual features of artworks are processed, just as with all other visual objects. Second, attentional processes redirect information processing to the prominent visual properties—such as color, shape, and composition—by means of the fronto-parietal attentional network. This process likely contributes to the vivid experience that characterizes VAE [4]. Third, the attentional networks modulate processing within the ventral visual stream, which leads to attributional networks. Specifically, the content of artworks (landscapes, portraits, etc.) is processed by the attributional areas of the ventral visual stream. Fourth, feedback/feedforward processes, linking attentional and attributional circuits in the ventral visual stream, enhance the experience of the visual objects. Finally, emotional systems mainly located in the frontal cortex and partially belonging to the so-called Default Mode Network (DMN; [5]) may be involved in VAE, especially when individuals explicitly focus on emotions raised from watching works of arts, such as horror, disgust, and so on [6]. In the past decade, several studies assessing the neural underpinnings of aesthetic experience, especially by using functional magnetic resonance imaging (fMRI) in healthy participants ([7,8] for review and meta-analysis), found evidence that strongly supports this model of VAE. Indeed, VAE has been found to rely on a wide network of brain areas consisting of sensory systems [6] and attributional areas of the ventral visual stream [6,7,9], differently engaged as a function of the content of the artwork [7]. Within this network, aesthetic preference (e.g., positive or negative aesthetic judgment or the degree of experiences attractiveness) is also processed [10,11]. Other regions—specifically, the orbito-frontal and medial-frontal cortex, anterior cingulate, and insula, [6,10,12,13]—have been found to process more emotional aspects [14,15]. 

Thus, VAE arises from a set of different cognitive, affective, and emotional processes and, accordingly, it is subtended by different brain networks. Neuropsychology of visual art may give some important insights into the effect of specific neuropsychological deficit on VAE. As for all other visual objects, perception of artworks relies on perceptual constancy, object size, and orientation on pictorial recognition through the matching of new visual inputs with previously acquired concepts, as well as on isolating a single figure from the context (i.e., disembedding), and it obeys the global precedence effects as it happens for all other visual objects [16]. Localized brain damage may selectively affect any or all of these components leading to specific neuropsychological deficits. For example, a lesion of the temporal lobe may lead to a deficit in object-size estimation, with objects appearing nearer or further away (respectively, pelopsia and teleopsia) [17]. A lesion of the ventral visual stream may lead to apperceptive agnosia [17], whereas a disconnection between visual and semantic brain regions may lead to associative agnosia [18], that is a deficit in object meaning with intact object perception. Also, a deficit in isolating figures in tangled pictorial array [19] may occur after lesion of the posterior brain regions. Furthermore, global precedence effect, that is, the domination of the whole spatial layout on the local features [20], may be affected by brain lesions. Specifically, patients with lesions of the left hemisphere typically depict global features but exclude the details whereas patients with lesions of the right hemisphere present the opposite pattern, that is, they depict details but exclude the global features [21,22,23]. 

Despite effects of unilateral brain damage and the neuropsychological deficits on established visual artists that have been widely described in literature [16,24], little is known about whether or not and how a lesion in the above mentioned neural network of areas (see [7] for review and meta-analysis) affects the VAE in the beholders. One previous noteworthy study has assessed neural correlates of the assessment of art attributes (i.e., both conceptual and perceptual attributes) in right brain damaged patients using Voxel Lesion Symptom Mapping (VLSM) [25] finding that damage of the right hemisphere due to stroke impaired the assessment of the conceptual (i.e., abstractness, accuracy) and perceptual attributes of the artworks. Specifically, different regions within the frontal, parietal and temporal lobes have been found to be related to the assessment of conceptual attributes (i.e., abstractness, symbolism, realism, and animacy) whereas the inferior prefrontal cortex has been found to affect perceptual attributes (i.e., depth). Importantly, patients did not differ from age-matched controls in their aesthetic preference. With Alzheimer’s Disease (AD), where the damage in the brain is diffuse, two studies found that aesthetic preference in patients with AD is stable over time. AD patients have been found to be consistent in the rating of visual artworks when tested twice within one [26] or two weeks [27], even if they failed in recognizing the paintings they rated [26,27]. Interestingly, aesthetic preference in AD has been found to be stable over two weeks for portrait paintings, landscape paintings, and landscape photographs, but not for portrait photographs [28] suggesting that artistic portraits are processed differently from natural faces in pathological aging. Also, aesthetic preference of patients with frontotemporal dementia remained stable over two weeks, despite severe language deficits [29].

Here, for the first time, we assessed whether unilateral left or right hemispheric lesions affect the aesthetic judgment—that is the judgment of pleasantness experienced by watching the painting—of the part/whole ambiguity in art. Ambiguity is a trick artists use to evoke an esthetic experience in the beholder. Part/whole ambiguity is the hallmark of Arcimboldo’s artworks. In these artworks, each ‘part’ (e.g., book) has a perceptual meaning per se and is combined with others in order to create a ‘whole’ configuration with a different meaning (i.e., a human face). Thus, more than one perceptual interpretation is possible at the same time and each painting could be interpreted in two ways, as an array of objects or as a face. Previously, with an fMRI study, aesthetic judgment of ambiguous paintings has been found to selectively involve the right superior parietal lobe, together with other brain areas usually involved in VAE, such as the orbitofrontal cortex and anterior cingulate cortex [10]. Furthermore, the aesthetic judgment of Arcimboldo’s ambiguous artworks was found to engage a specific mechanism within the fusiform face area (FFA), with artworks eliciting a negative judgment leading to more pronounced activation of the FFA than ambiguous artworks eliciting a positive judgment [10]. Behavioral investigations found that individual local-global perceptual style selectively affected VAE (i.e., aesthetic and ambiguity judgments) of Arcimboldo’s paintings whereas it did not affect the VAE of renaissance portraits [30]. Specifically, individuals with local perceptual style rather than judging Arcimboldo’s as more ambiguous then individuals with global perceptual style, preferred more these ambiguous portraits than the latter [30]. This result, taken together with neuroimaging evidence [10], suggests that VAE of Arcimboldo’s Portraits is connected more with the local processing of the object than with the global processing of the face. Global/local processing, as well as visuo-spatial and visuo-constructional abilities within the right parietal cortex have been posited as pivotal for VAE [16]. Arcimboldo’s ambiguous collection, by enhancing the effect of visuo-spatial and visuo-perceptual abilities, may be a good tool to investigate the effect of brain lesions and neuropsychological functioning on VAE. We expected that the damage of the right hemisphere, by affecting visuo-spatial and visuo-perceptual abilities, may yield a difference in the aesthetic judgment of these artworks and not of other kinds of artworks (i.e., renaissance portraits). We also tested the correlation between the aesthetic judgment of ambiguous and non-ambiguous portraits and the visuo-spatial, visuo-perceptual, and constructional abilities.

To test our hypotheses, we asked 19 unilaterally brain-damaged patients (10 LBDP and 9 RBDP) and 20 healthy individuals (C), matched for age and education with the patients (Mean Age of the sample = 59.00, SD = 12.50; Mean Education of the sample = 10.56; SD = 3.31; see the method section below for more information about the demographics of LBDP, RBDP, and C), to rate on a Visual Analogue Scale (VAS) the pleasantness of 10 of Arcimboldo’s portraits (AP), 10 renaissance portraits (RP), 10 still life paintings (SL), and 10 of Arcimboldo’s modified portraits, where only objects/parts are detectable (AO). All individuals also performed a Navon task and a facial recognition test; patients were also administered an extensive neuropsychological evaluation of visuo-perceptual and visuo-constructional abilities.

## 2. Results

Figure 1 shows the lesion reconstruction of 9 (4 RBDB and 5 LBDP) out of 19 patients. Note that this imaging procedure and data had a descriptive aim, not a quantitative or statistical one (see also the method section below for further information).

The MANOVA on mean VAS scores showed a main effect of Group [*F* (2, 36) = 4.56; *p* = 0.02, *η_p_*^2^ = 0.20; Observed Power = 0.74] on aesthetic judgment of AP. Post-hoc comparisons showed that RBDP liked AP more than C (*p* = 0.02) (Table 1). No significant differences between groups have been observed for RP [*F* (2, 36) = 0.96; *p* = 0.39, *η_p_*^2^ = 0.05; Observed Power = 0.20], AO [*F* (2, 36) = 0.87; *p* = 0.43, *η_p_*^2^ = 0.05; Observed Power = 0.19] and SL [*F* (2, 36) = .13; *p* = 0.88, *η_p_*^2^ = 0.01; Observed Power = 0.07]. 

The MANOVA on the accuracy of face perception in AP and RP did not reveal any significant difference among groups [AP: *F* (2, 36) = 1.32; *p* = 0.28, *η_p_*^2^ = 0.07; Observed Power = 0.27; RP: *F* (2, 36) = 0.004; *p* = 0.996, *η_p_*^2^ = 0.00; Observed Power = 0.05] as well as MANOVA on the accuracy on Local [*F* (2, 36) = 0.64; *p* = 0.54, *η_p_*^2^ = 0.03; Observed Power = 0.15] and Global [*F* (2, 36) = 0.86; *p* = 0.43, *η_p_*^2^ = 0.05; Observed Power = 0.19] level of the Navon Task.

Table 2 shows Spearman’s correlation coefficients. We found that Aesthetic Judgment of AP was significantly positively correlated with Object Decision subtest of VOSP. Interestingly, Aesthetic Judgment of AP was also correlated with Aesthetic Judgment of AO, but not with RP and SL. Otherwise, Aesthetic Judgment of RP, SL, and AO were positively correlated. Furthermore, Aesthetic Judgment of RP, SL, and AO were not correlated with any of the investigated visuo-perceptual and constructional abilities.

A set of analyses was also performed to analyze individual performances. The results of the Crawford’s analysis are summarized in Table 3. Two patients in the RBDP group [Pt 7: t(1, 19) = 3.61, *p* < 0.01; Pt 8: t(1, 19) = 2.52, *p* = 0.02] and two patients in the LBDP group [Pt 17: t(1, 19) = 3.49, *p* < 0.01; Pt 19: t(1, 19) = 2.19, *p* = 0.04] liked AP more than C. Interestingly, none of these patients were affected by neuropsychological deficit in any of the investigated visuo-perceptual and visuo-spatial functions.

## 3. Discussion

Our aim was to investigate whether left or right hemispheric lesions affect the aesthetic judgment of the part/whole ambiguity in visual art. There is an advantage in using Arcimboldo’s paintings as stimuli in aesthetic judgment studies, namely they control for artistic effects within the same artist and within the same painting, that is, the same artist’s cognition, artistry, talent, and creativity are represented in a single painting. Furthermore, they allow for studies of the effect of part/whole ambiguity in aesthetic judgment, which have been found to rely on specific neural networks [10] and perceptual skills [30]. Thus, the study of ambiguity in artworks is a useful tool in the cognitive neuroscience of art, especially with regards to the assessment of the visuo-perceptual and visuo-spatial processes underlying on VAE of complex artworks. 

With Arcimboldo’s paintings, we found that unilateral brain damage of the right hemisphere is related to a higher rating (i.e., the patients liked the paintings), while the rating of the other artwork categories, SL and RP was not affected by the laterality of the brain damage. On the one hand, this result confirms that ambiguity is a useful condition to assess neural underpinning of VAE [31]. On the other hand, the result suggests that the effect of brain damage in aesthetic judgment may be specific for complex artworks, such as those showing part/whole ambiguity. Finding that damage in the right hemisphere, but not the one in the left hemisphere, affects aesthetic judgment of AP suggests a pivotal role of the right hemisphere. This result is consistent with a previous neuroimaging investigation, which found that the right parietal lobe is engaged during the aesthetic judgment of ambiguous painting [10] and extends previous literature on brain asymmetries in aesthetic experience and art appreciation [32,33,34,35] to aesthetic appreciation of ambiguity in artworks.

An interesting result of the present study is the significant positive correlation between aesthetic judgment of AP and the performances on Object Decision subtest of the VOSP. In the latter test, individuals are required to choose which one of four silhouettes representing the shadow of a real object. As stated in the introduction section, pictorial recognition is one of the stages characterizing neuropsychology of VAE. Our results strongly support the idea that pictorial recognition may affect the aesthetic judgment of the beholders in the case of complex artworks, such as AP. Interestingly, a previous study found that RBDP’s performances on shape detection task of the VOSP (i.e., the preliminary step of the VOSP aimed to assess minimal visual and sensorial skills) were correlated with the assessment of simplicity in artworks [25]. We failed to find a significant effect due to the local/global processing assessed by means of the Navon task. This is likely due to the use of a less sensitive measure such as accuracy. Actually, we did not analyze patients’ reaction times on the Navon task because performance is sensitive to motor impairments, particularly in neurological patients. In other studies, reaction times on the Navon task have been found to be a good tool to assess individual preference towards Global or Local level of percept and its effect on aesthetic judgment of AP [30]. This limitation makes impossible to draw definite conclusion about the effect of local/global processing in our sample and suggests the use of a more fine-graded index in further investigations. Interestingly, aesthetic judgment of AP was also correlated with aesthetic judgment of AO, but not with that of RP and SL. Also, aesthetic judgment of RP, SL, and AO were positively correlated, but none of them was correlated with any of the investigated visuo-perceptual and constructional abilities. This suggests that aesthetic judgment of AP significantly differs from that of non-ambiguous artworks and represents a good tool to assess neuropsychological dimensions of visual aesthetic experience of the beholders. This is probably due to the part/whole ambiguity that characterizes AP artworks and that inevitably strains visuo-perceptual abilities. This idea is strongly supported by the correlation between the aesthetic judgment of AP and performances on the object decision subtest of the VOSP. Further investigations are needed to explore the possible ‘causative role’ of a constructional and perceptual deficit on VAE of non-ambiguous complex artworks, such as landscapes.

We further explored the possibility that abnormal aesthetic judgment of AP was due to the specific neuropsychological profile in the above-mentioned abilities. However, we failed to find specific dissociation due to the deficit and aesthetic judgment in our sample. It should be stressed that single case analysis, even if suitable in the case of small sample to clarify the effect of specific neuropsychological deficit, may be poorly informative in the case of aesthetic judgment because it does not allow for controlling the subjective aesthetic preference. In other words, they could be more informative about the individual’s preference than the neuropsychological functioning we were interested in. 

A final consideration refers to the neural correlates of the aesthetic judgment of ambiguous artworks. We found that right hemispheric lesions, but not left hemispheric lesions, alter the rating of Arcimboldo’s portraits. This is the very first time that a positive relationship between right hemisphere lesions and aesthetic judgment of ambiguity in artworks has been demonstrated. As described in the introduction section, in drawing tasks RBDP commonly draw the details of objects but exclude the global features of the objects [21,22,23]. Based on previous behavioral and neuroimaging findings [10,30], suggesting that aesthetic judgment of AP relies on the appreciation of the parts more than the whole, we may speculate that, even in absence of blatant and conspicuous exploration deficit and CA (see Tables 3 and 4) in our patients with right hemisphere damage appreciation of the local/parts in Arcimboldo’s paintings contributed to the aesthetic judgment. Anyway, further investigations are needed to test such a hypothesis. The specific region and network within the right hemisphere remains to be determined, even if descriptive data about lesion location (Table 3) seem to suggest a pivotal role of the parietal lobe, which is consistent with a previous neuroimaging investigation on aesthetic judgment of ambiguous paintings in healthy participants [10]. Future studies are needed to obtain a clearer localization of the aesthetic judgment of ambiguous artworks, for example by using voxel lesion symptom mapping (VLSM).

## 4. Materials and Methods

### 4.1. Participants and Neuropsychological Evaluation

Participants included 19 brain-damaged patients and 20 healthy individuals with no history of neurological or psychiatric impairment (controls; C) (see Table 4 for details). The study was designed in accordance with the ethical principles of the Declaration of Helsinki and the protocol was approved by the local ethical committee, and written informed consent was obtained from each participant before starting the study.

All patients were recruited from a population of in-patients at the IRCCS Fondazione Santa Lucia in Rome (Italy) who were being treated for hemiparesis or hemiplegia following cerebrovascular accident (CVA). Exclusion criteria were the presence of two or more CVA, neoplastic or traumatic etiology, and cognitive deterioration. C were relatives of in-patients at the IRCCS Fondazione Santa Lucia, who participated to the study as volunteers.

The brain-damaged patients were subdivided according to lesion side: 10 patients had a unilateral left brain lesion (left brain damaged patients, LBDP) and 9 patients had a unilateral right lesion (right brain damaged patients, RBDP). None of the LBDP were affected by a severe comprehension deficit or global aphasia and none of the RBDP showed perceptual neglect (see below the Neuropsychological Examination Section for more details).

Age, *F* (2, 36) = 00.02; *p* = 0.98; education, *F* (2, 36) = 1.15; *p* = 0.33 and time from stroke onset, t(17) = −0.88; *p* = 0.39 did not differ among groups (Table 4).

### 4.2. Lesion Reconstruction

Brain lesions were manually reconstructed from magnetic resonance imaging (MRI) or computed tomography (CT) scans onto a standard template using MRIcron [36]. Lesion reconstruction was possible for 9 (4 RBDB and 5 LBDP) out of 22 patients. Note that this imaging procedure and data had a descriptive aim, not a quantitative or statistical one. With this descriptive aim, we provide the lesion overlap of RBDP (Figure 1A) and LBDP (Figure 1B).

### 4.3. Neuropsychological Investigation

To exclude deficits in abstract reasoning abilities, all patients and C have been also assessed with the *Raven’s Colored Progressive Matrices* (RCPM; [37]). Participants whose age was above 65 years were also administered the *Mini-Mental State Examination* (MMSE; [38]). None of the participants performed below the cut-off in these tasks. 

RBDP and LBDP underwent an exhaustive neuropsychological evaluation as function of lesion side, to exclude the presence of perceptual neglect (RBDP) and comprehension deficit (LBDP). RBDP have been assessed for both extrapersonal and personal neglect. Extrapersonal neglect has been assessed by using the *Standard Battery for the Evaluation of Hemineglect* [39]. This battery includes four tests: Letter Cancellation Test, Line Cancellation Test, Wundt–Jastrow Area Illusion Test, and Sentence Reading Test. Patients are diagnosed as having visuospatial neglect if they perform below the cutoff on at least two of the four tests. Personal hemi-neglect of space has been assessed using the Use of Common Objects Test [40], which requires using three objects (eyeglasses, a razor, or face powder and a comb) in the body space. A diagnosis of personal neglect was made if the total score in the Use of Common Objects test was greater than or equal to 2 [41] (Zoccolotti, Antonucci, & Judica, 1992). None of the RBDP of the present study performed below the cut-off on any of these tests. Comprehension deficit in LBDP has been assessed by using the Token Test [42] or comprehension subtests of the Battery for the Analysis of Aphasic Deficit [43] or Neuropsychological Exam for Aphasia [44]. None of the LBDP in this study performed below the Italian cut-off. 

All RBDP and LBDP underwent an exhaustive examination of visuo-constructional and visuo-perceptual abilities. Constructional apraxia (CA) has been assessed by asking them to copy geometrical figures [42]. Visuo-perceptual abilities have been explored by using the Incomplete Letter and the Object Decision subtests of the Visual Object and Space Perception Battery (VOSP; [45]). Apperceptive visual agnosia has been further assessed by Overlapping Figure Test [46]. Prosopagnosia has been assessed by using the short version of the Benton Facial Recognition Test (BFRT; [47]). No below-normal performance has been recorded on these tests in this group of patients.

### 4.4. Experimental Procedures

#### 4.4.1. Aesthetic Judgment: Visual Analogue Scale

Stimuli: Our set of stimuli included (1) 10 Arcimboldo’s ambiguous portraits (Arcimboldo’s portraits, AP); (2) 10 realistic Renaissance portraits (RP), which were matched with Arcimboldo’s portraits for gender of the portrayed sitter and face position and (3) 10 still life paintings (SL). We also modified 10 Arcimboldo’s ambiguous portraits in order to hidden the face. Hereafter, we refer to these stimuli as Arcimboldo’s objects (AO). APs and RPs have been extracted from the set of stimuli used in Boccia et al. [10,30]. The stimuli measured 12.5 × 18.4 cm (width × height) and were presented on cardboards.

Procedure: Participants sat in front of the experimenter. The experimenter showed artwork depicted on cardboards one-by-one. Participants were asked to rate the artworks according to the pleasantness they experienced by watching them, using a visual analogue scale (VAS) of 10 cm. For each participant and condition (i.e., AP, RP, SL, AO) we calculated the mean VAS score (i.e., rating).

#### 4.4.2. Face Perception

Stimuli: The same set of stimuli of aesthetic judgment has been used in a computerized task aimed at assessing whether participants correctly perceive a face in AP and RP. The stimuli measured 500 × 750 pixels (width × height) and were projected in the center of the screen.

Procedure: Participants were seated in front of a computer screen (PC laptop with a Windows 7 operating system). They were asked to press the green button on the keypad if the stimuli represented a face (target condition) and to press the red button if other stimuli (e.g., objects) appeared. The stimuli, presented in a randomized order, remained on the screen until one of the two response keys had been pressed. For each participant, we calculated the accuracy as the sum of the correct identified faces in AP and RP. 

#### 4.4.3. Global/Local Perception: The Navon Task 

We used an adapted version [30] of the classical Navon task [20,48], to assess participants’ global or local perception.

Stimuli: The target condition consisted of global letters that were either Es composed of local Hs, Ls, or Ts, or global Hs, Ls, or Ts composed of local Es. In the nontarget condition, the global letters were either Fs composed of local Hs, Ls, or Ts or global Hs, Ls, or Ts composed of local Fs. The global letters measured 250 × 400 pixels (width × height) and were presented in a randomized order at the center of the computer screen.

Procedure: Participants were asked to press the green button on the keypad every time the letter E appeared at either the global or the local level (target condition) and to press the red button if other stimuli appeared on the screen. The stimuli remained on the screen until one of the two response keys had been pressed. For each participant and condition (i.e., global and local targets), we calculated the accuracy as the sum of correct responses.

### 4.5. Data Analysis

We performed a (1) Multivariate Analysis of Variance (MANOVA) on values of Aesthetic Judgment obtained using VAS, by entering the Group (C vs. RBDP vs. LBDP) as between factor and the mean rating on the VAS on AP, AO, RP, and SL as dependent variables; MANOVAs with Group as between factor have been also performed on (2) the accuracy of face perception in AP and RP and (3) the accuracy on the Global and Local levels of the Navon task. Post-hoc comparisons between groups have been performed by using Bonferroni’s adjustment for multiple comparisons; (4) Finally, we performed an explorative non-parametric correlation analysis (Spearman’s Rho) between the mean rating on the VAS on AP, RP, AO, and SL, and the performances on visuo-constructional and visuo-perceptual tests of RBDP and LBDP.

We also performed Crawford’s analysis using SINGLIMS_ES.exe [49,50] to determine whether the rating on the VAS at AP of individual patients were significantly lower/higher than those of Cs. Thus, each individual is treated as a sample of *N* = 1. This analysis is the most suitable statistical approach when the normative sample is small [51].

## 5. Conclusions

In conclusion, our results suggest that even in the absence of exploration or constructional deficits, lesions of the right hemisphere affect the aesthetic judgment of ambiguous artworks (i.e., Arcimboldo’s portraits), which is different from aesthetic judgment of other class of artworks investigated here (still life paintings and renaissance portraits). Our conclusions are limited to the aesthetic judgment of artistic portraits and objects. We did not assess other possible dimensions of VAE, such as the judgment of conceptual and perceptual attributes, nor the emotional engagement. Thus, more general conclusions about the VAE of ambiguous artworks in brain damaged patients remain to be determined in future studies. 

## Figures and Tables

**Figure 1 behavsci-07-00013-f001:**
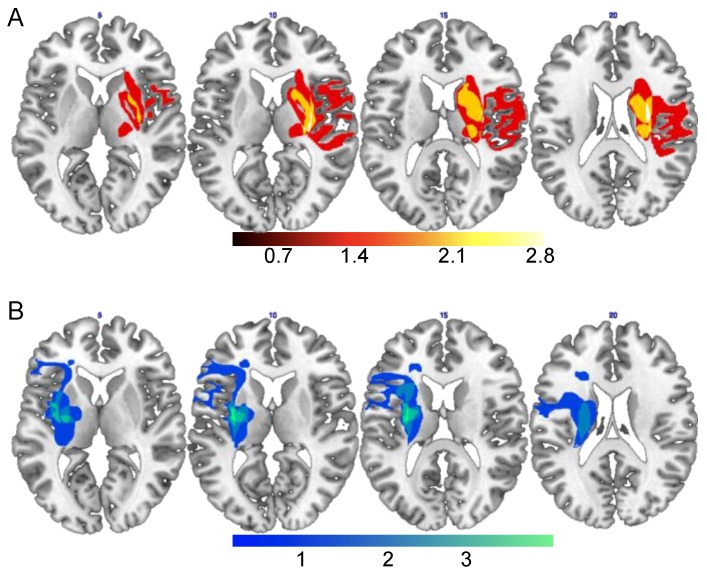
Regions of lesion overlap of RBDP (**A**) and LBDP (**B**). *RBDP’ lesion overlap* included precentral and postcentral gyri, superior and inferior frontal sulci/gyri, in the lateral and medial brain surfaces, supplementary motor area, insula, anterior and middle cingulate cortex, supramarginal gyrus, caudate nucleus, putamen, pallidum and thalamus, and superior temporal gyrus. *LBDP’ lesion overlap* included postcentral gyrus, superior and inferior frontal gyri/sulci, insula, hippocampus and parahippocampal cortex, amygdala, supramarginal gyrus, caudate nucleus, putamen, pallidum, thalamus, and superior temporal gyrus.

**Table 1 behavsci-07-00013-t001:** Mean and Standard Deviation on experimental tasks. Significant differences are marked with an asterisk (*). Notes. AP = Arcimboldo’s portraits; AO = Arcimboldo’s objects; RP = Renaissance portraits; SL = Still life.

Experimental Condition	C	RBDP	LBDP
*Rating on the VAS*			
AP	2.45	4.82 *	3.61
	(1.62)	(2.25)	(2.41)
AO	4.60	5.61	5.32
	(2.11)	(2.22)	(1.88)
RP	5.36	6.48	6.01
	(2.19)	(2.32)	(2.18)
SL	5.59	5.70	6.05
	(2.27)	(2.58)	(2.10)
*Faces perception*			
AP	9.85	9.33	9.70
	(0.37)	(1.41)	(0.67)
RP	9.90	9.89	9.90
	(0.31)	(0.33)	(0.32)
*Navon task*			
Global	6.85	7.22	4.40
	(5.71)	(5.38)	(4.55)
Local	10.40	11.67	10.80
	(3.69)	(0.50)	(1.55)

**Table 2 behavsci-07-00013-t002:** Spearman’s correlation coefficients. Significant correlations are marked with asterisks (* *p* < 0.05; ** *p* < 0.01). Notes. BFRT = Benton Facial Recognition Test; AP = Arcimboldo’s portraits; AO = Arcimboldo’s objects; RP = Renaissance portraits; SL = Still life.

Task	AP Pleasantness	AO Pleasantness	RP Pleasantness	SL Pleasantness	BFRT	Incomplete Letter	Object Decision	Overlapping Figure Test	Constructional Apraxia
AP Pleasantness	1.00	0.59 **	0.39	0.15	0.15	0.42	0.50 *	0.40	0.32
AO Pleasantness		1.00	0.66 **	0.56 *	0.06	0.09	0.01	0.04	−0.13
RP Pleasantness			1.00	0.57 *	−0.21	−0.11	−0.12	−0.09	−0.05
SL Pleasantness				1.00	0.24	−0.20	−0.28	−0.14	−0.17
BFRT					1.00	0.19	0.47	0.36	0.14
Incomplete Letter						1.00	0.46	0.10	0.25
Object Decision							1.00	0.56 *	0.31
Overlapping figure test								1.00	0.21
Constructional Apraxia									1.00

**Table 3 behavsci-07-00013-t003:** For each patient the lesion site, the performances at visuo-constructional and visuo-perceptual tests, the mean rating on the VAS on AP, and the results of the Crawford’s analysis are reported. Patients who showed significant difference on AP are reported in italics. Notes. * deficit; na, not applicable; ^a^ Cut-off of the Overlapping figure test is 1 error, because none of the Cs make any error. F = frontal lobe; O = occipital lobe; P = parietal lobe; T = temporal lobe; I = insula; c = cortical; sc = subcortical; th = thalamus; C = capsula; iC = internal capsula; ln = lenticular nucleus; bg = basal ganglia; p = posterior; a = anterior; bb = bulbo; crb = cerebellum; HC = hippocampus; paI = parainsular cortex; cr = corona radiate; cp = cerebral peduncle.

Patients	Lesion Site	BFRT	CA	Incomplete Letter	Object Decision	Overlapping Figure Test	AP	T(1, 19)	*p*
RBDP									
Pt 1	F, I	36 *	7 *	na	10 *	20 *^,a^	4.63	1.31	0.21
Pt 2	F-sc	53	+	18	18	25	5.11	1.60	0.13
Pt 3	T, O, P	43	14	20	15	22 *^,a^	0.43	−1.22	0.24
Pt 4	Th, C	45	14	20	18	25	5.66	1.93	0.07
Pt 5	T-sc, ln, bg	45	9	19	16	24 *^,a^	5.09	1.59	0.13
Pt 6	F, T, bg (c/sc)	44	12	19	19	25	4.53	1.25	0.23
*Pt 7*	*pP (c/sc)*	*45*	*14*	*20*	*18*	*25*	*8.44*	*3.61*	*0.00*
*Pt 8*	*F, P, T*	*43*	*14*	*20*	*19*	*25*	*6.63*	*2.52*	*0.02*
Pt 9	F, P, T	48	14	19	14	25	2.86	0.25	0.81
LBDP									
Pt 10	F, P	47	7 *	20	18	25	1.72	−0.44	0.67
Pt 11	ln, C	51	12	18	19	25	3.02	0.34	0.74
Pt 12	bb	49	9	19	15	24 *^,a^	2.66	0.13	0.90
Pt 13	ln, C	34 *	7 *	15 *	14	25	0.30	−1.30	0.21
Pt 14	F, T, I (c/sc)	42	14	17	18	25	5.81	2.02	0.06
Pt 15	Th, C, crb	36 *	14	18	15	24 *^,a^	2.28	−0.10	0.92
Pt 16	F, T (c/sc)	47	13	17	11 *	25	3.87	0.86	0.40
*Pt 17*	*T-HC, paI, bg-iC, cr*	*38*	*9*	*20*	*16*	*25*	*8.25*	*3.49*	*0.00*
Pt 18	Th, C, T, P, cr	38	9	15 *	13 *	24 *^,a^	2.10	−0.21	0.84
*Pt 19*	*F, P, iC, cp*	*50*	*13*	*20*	*18*	*25*	*6.09*	*2.19*	*0.04*
						Controls			
						Mean	2.45		
						S.D.	1.62		

**Table 4 behavsci-07-00013-t004:** Demographics and neuropsychological data. Notes. RCPM = Raven’s Colored Progressive Matrices; BFRT = Benton Facial Recognition Test; VOSP = Visual Object and Space Perception Battery.

Demographics/Neuropsychological Tests	C	RBDP	LBDP
Age	59.30	58.22	59.10
	(10.54)	(14.83)	(15.14)
Education	11.05	11.00	9.20
	(3.30)	(3.12)	(3.43)
Time from Stroke (Days)	-	42.11	53.60
	-	(13.52)	(36.73)
RCPM	32.8	25.67	25.70
	(2.97)	(5.72)	(4.85)
BFRT	-	44.67	43.20
	-	(4.50)	(6.34)
Incomplete Letter (VOSP)	-	19.38	17.90
	-	(0.74)	(1.91)
Object Decision (VOSP)	-	17.75	15.90
	-	(1.49)	(2.85)
Overlapping figure test	-	24.00	24.70
	-	(1.80)	(0.48)
Constructional apraxia	-	12.25	10.70
	-	(2.76)	(2.79)

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
