# Peer review of "Neuropsychology of Aesthetic Judgment of Ambiguous and Non-Ambiguous Artworks"

_behavsci, 2017, doi:10.3390/bs7010013_

Round 1

Reviewer 1 Report

This manuscript has strenghts: it is well organized and clearly written.

Minor revisions:

1.      Line 49: the phrase “in most cases, emotional systems are involved” is too vague. The authors could specify which are the cases and anticipate the emotional systems involved set out in lines 56 and 57.

2.      Line 86: The authors could better expain the results of the two studies are about aesthetic experience. Moreover, the word “experience” is used as a synonym for “preference” but I am not sure it is right.

3.      Line 105: authors could add information about the sample: for example the mean age of the whole sample. Moreover they can add the reference to Table 4.

4.      Line 197: How participants have been collected? Which was the method of recruitment? Participants were paid?

Author Response

Reviewer 1 (R1):

1)    R1: This manuscript has strengths: it is well organized and clearly written.

Authors’ reply: We would like to thank the Reviewer for his/her positive comments and useful suggestions, we now implemented in the revised version of the manuscript.

2)    R1: Line 49: the phrase “in most cases, emotional systems are involved” is too vague. The authors could specify which are the cases and anticipate the emotional systems involved set out in lines 56 and 57.

Authors’ reply: We have better specified which are these cases and anticipated the neural systems reported afterwards in the text.

3)    R1: Line 86: The authors could better explain the results of the two studies are about aesthetic experience. Moreover, the word “experience” is used as a synonym for “preference” but I am not sure it is right.

Authors’ reply: We have explained better the results of these two studies and replaced the word “experience” with “preference”.

4)    R1: Line 105: authors could add information about the sample: for example the mean age of the whole sample. Moreover they can add the reference to Table 4. 

Authors’ reply: We have added the mean age and education (and standard deviations) of the whole sample in lines 114-116. We have also included a reference to the method section, where the table 4 is also listed according to the Journal Guidelines, which requested to include the method section at the end of the manuscript.

5)    R1: Line 197: How participants have been collected? Which was the method of recruitment? Participants were paid?

Authors’ reply: Participants have been recruited at the IRCCS Fondazione Santa Lucia. C were relatives of in-patients at the IRCCS who volunteered to the study (they were not paid). We have added this information in the revised version of the manuscript.

Reviewer 2 Report

Review of manuscript entitled “neuropsychology of aesthetic reactions to ambiguous and non-ambiguous artworks”

This paper presents a study of aesthetic appreciation by left- and right- brain lesion patients. The authors report that participants with lesions in their right hemisphere expressed greater aesthetic pleasure towards ambiguous artworks than control participants. This is a hard paper to evaluate. On the one hand, the aims are interesting and it is clear that the authors have invested a great deal of time and effort to collect the data. On the other hand, the many conceptual and methodological problems lead me to believe that this work cannot contribute to further our understanding of the neural foundations of aesthetic experience.

Conceptual problems:

1. What is the target of the study? The title refers to “aesthetic reactions”, the abstract to “aesthetic pleasure”, the paper to “aesthetic visual experience”, “liking”, “aesthetic appreciation”, “aesthetic assessment”, etc. It seems to me that the authors have not appropriately identified the phenomenon they are attempting to study.

2. The authors (line 34) define aesthetic experience as “the states of the mind and brain during the perception of works of art”. Does this mean that one cannot have an aesthetic experience to nature? Or to faces or bodies of other people? Does my appreciation of the beauty of nature or my partner not count as aesthetic? The authors conflate aesthetic experience with the appreciation of art. Please see Pearce et al. (2016) or other sources to clarify this issue.

3. The authors (line 55) equate the “valence” of aesthetic experience with “like” or “dislike”. Not only does this seem arbitrary, it goes against much research showing that liking and disliking are related to the arousal dimension of affect (Berlyne, 1971; Marin & Leder, 2016).

4. The paper reports the effects of brain damage on the aesthetic appreciation of artworks, to use Pearce and colleagues’ (2016) terms. But the introduction overlooks much of the most significant work on the topic, especially the paper by Bromberger, Sternschein, Widick, Smith II and Chatterjee (2011). The authors do refer to two papers on the effects of neurodegenerative diseases, but they overlook others (e.g., Graham, Stockinger, & Leder, 2013; Halpern & O’Connor, 2013). It is also surprising, given the authors’ interest in the effects of lesions in left and right hemispheres, that they overlook the extensive literature on brain asymmetries in aesthetic experience and art appreciation (e.g., Beaumont, 1985; Coney & Bruce, 2004; Ellis & Miller, 1981; Levy, 1976; Zaidel, 1994, 2013, 2015; Zaidel & Kasher, 1989). A thorough coverage of the literature is good scientific practice, helps ground the study’s aims, and allows the proper assessment of its contribution.

Methodological problems:

5. The authors asked participants to rate the stimuli’s pleasantness. I would not say that this is an aesthetic assessment. It is an assessment of the valence dimension of the paintings. Despite the authors’ interchangeable use of this term with “aesthetic reaction”, “liking”, “aesthetic experience”, and so on, they are not synonymous. Affective valence is surely an important contributor to aesthetic experience, but it isn’t aesthetic experience. This is captured in prominent models of aesthetic experience (Chatterjee, 2003; Leder, Belke, Oeberst, & Augustin, 2004), where affective processes are components of a larger scheme. Thus, the conclusions from this paper refer not to aesthetic reactions or to visual aesthetic experience, but to affective evaluation of artworks. To present them otherwise is misleading.

6. The authors provide the lesion reconstruction for only 9 of the 19 patients. I see this as the most damaging problems of all. In a nutshell, the authors cannot determine the precise brain region responsible for the effect they report, and therefore, they cannot provide the explanatory mechanism for it. The fact is that RBDP patients gave higher pleasantness ratings than control participants (they did not express higher aesthetic appreciation, as claimed by the authors in line 142). Why is this the case? Is this due to damage to brain regions involved in perception? Judgment? Affective processing? There is no way to tell because the location of the lesion is not appropriately described. The authors can only speculate, and do so in a most arbitrary way: “in our patients with right hemisphere damage appreciation of he local/parts in Arcimboldo’s paintings contributed to the aesthetic assessment”. Again, this is such a misleading statement that it borders on scientific fraud. First, participants were assessing the valence of the artworks. Second, the authors themselves report no difference between RDBP and controls in the local or global levels of the Navon task. In the final paragraph of the conclusions, the authors overlook the fact that their main argument is based on speculation and present it as suggested by the results: “our results suggest that even in the absence of exploration or constructional deficits, lesions of the right hemisphere, somehow prevent the processing of the global/whole feature of the paintings …” There is nothing in the results that suggest this.

As a consequence of all the above, it is unclear what the scientific value of this study is. The authors themselves observe that “further studies are needed to obtain a clear localization of VAE, for example, by using voxel lesion symptom mapping”. This sentence is quite striking because this is precisely what Bromberger et al. (2011) published 6 years ago. In their study they had a precise mapping of right hemisphere lesions, which they were able to relate using voxel lesion symptom mapping to specific deficits in processing different kinds of attributes of the artworks. Science needs to be a cumulative enterprise. Ignoring previous work, such as Bromberg and colleagues’ (2011), can only lead us in circles. The authors should have designed their study based on the existing knowledge, trying to refine it, or even question it if they wish. Because the shortcomings of the paper under review here were solved 6 years ago, and the conclusions are based solely on speculation and disregarding part of the results, it seems to me that this work cannot make any meaningful contribution to our knowledge of the neural foundations of the appreciation of art.

Minor issue: lines 81-83 please rephrase. This sentence is very confusing. It seems that the authors are suggesting that lesions in those brain regions lead to visual aesthetic experiences.

References

Beaumont, G. J. (1985). Lateral organization and aesthetic preference: The importance of peripheral visual asymmetries. Neuropsychologia, 23, 103-113.

Berlyne, D. E. (1971). Aesthetics and Psychobiology. New York: Appleton-Century-Crofts.

Bromberger, B., Sternschein, R., Widick, P., Smith II, W., & Chatterjee, A. (2011). The right hemisphere in esthetic perception. Frontiers in Human Neuroscience, 5, 109. doi: 110.3389/fnhum.2011.00109.

Chatterjee, A. (2003). Prospects for a Cognitive Neuroscience of Visual Aesthetics. Bulletin of Psychology of the Arts, 4, 55-60.

Coney, J., & Bruce, C. (2004). Hemispheric processes in the perception of art. Empirical Studies of the Arts, 22, 181-200.

Ellis, A. E., & Miller, D. (1981). Left and wrong in adverts: Neuropsychological correlates of aesthetic preference. British Journal of Psychology, 72, 225-229.

Graham, D. J., Stockinger, S., & Leder, H. (2013). An island of stability: art images and natural scenes – but not natural faces – show consistent esthetic response in Alzheimer’s-related dementia. Frontiers in psychology, 4, 107. doi:doi:10.3389/fpsyg.2013.00107

Halpern, A. R., & O’Connor, M. G. (2013). Stability of Art Preference in Frontotemporal Dementia. Psychology of Aesthetics, Creativity, and the Arts, 7, 95–99.

Leder, H., Belke, B., Oeberst, A., & Augustin, D. (2004). A model of aesthetic appreciation and aesthetic judgments. British Journal of Psychology, 95, 489-508.

Levy, J. (1976). Lateral dominance and aesthetic preference. Neuropsychologia, 14, 431-445.

Marin, M. M., & Leder, H. (2016). Effects of presentation duration on measures of complexity in affective environmental scenes and representational paintings. Acta Psychologica, 163, 38–58.

Pearce, M. T., Zaidel, D. W., Vartanian, O., Skov, M., Leder, H., Chatterjee, A., & Nadal, M. (2016). Neuroaesthetics: The cognitive neuroscience of aesthetic experience. Perspectives on Psychological Science, 11, 265-279.

Zaidel, D. W. (1994). Words apart: Pictorial semantics in the left and right cerebral hemispheres. Current Directions in Psychological Science, 3, 5-8.

Zaidel, D. W. (2013). Split-brain, the right hemisphere, and art: Fact and fiction. Progress in Brain Research, 204, 3-17.

Zaidel, D. W. (2015). Hemispheric specialization, art, and aesthetics. In J. P. Huston, M. Nadal, F. Mora, L. F. Agnati, & C. J. Cela-Conde (Eds.), Art, aesthetics and the brain (pp. 373-382). Oxford: Oxford University Press.

Zaidel, D. W., & Kasher, A. (1989). Hemispheric memory for surrealist versus realistic paintings. Cortex, 25, 617-641.

Author Response

Reviewer 2 (R2):

1)    R2: This paper presents a study of aesthetic appreciation by left- and right- brain lesion patients. The authors report that participants with lesions in their right hemisphere expressed greater aesthetic pleasure towards ambiguous artworks than control participants. This is a hard paper to evaluate. On the one hand, the aims are interesting and it is clear that the authors have invested a great deal of time and effort to collect the data. On the other hand, the many conceptual and methodological problems lead me to believe that this work cannot contribute to further our understanding of the neural foundations of aesthetic experience.

Authors’ reply: We appreciate his/her review since it enlightens the need to clarify the aims and the limits of present study and to use a more precise language (see the points below). We have extensively revised the manuscript according to the Reviewer’s criticisms, also expanding upon the aims and the limitations of the current investigation.

2)    R2: What is the target of the study? The title refers to “aesthetic reactions”, the abstract to “aesthetic pleasure”, the paper to “aesthetic visual experience”, “liking”, “aesthetic appreciation”, “aesthetic assessment”, etc. It seems to me that the authors have not appropriately identified the phenomenon they are attempting to study.

Authors’ reply: We thank the Reviewer for raising this issue that allows as to ameliorate the manuscript. Here, we were interested in studying if left or right hemispheric lesions affect the possibility to positively appreciate the effect of the use of the part/whole ambiguity in art, that is a never investigated issue in the field of the neuroaesthetic. We would like to highlight that we refer to the Visual Aesthetic Experience (VAE) as a whole when we are referring to previous studies assessing (or modelling) Aesthetic Experience as a whole (see for example lines 39/40) or when different aspects of Aesthetic experience have been investigated (for example our previous study assessing ambiguity evaluation and aesthetic reaction to Arcimboldo’s). We recognize the need to be more consistent in the term used to identify the target of the “current” study. We have provided an operational definition of the target of the study, that is the Aesthetic Judgment, at the end of the introduction section. We consistently use the definition “aesthetic judgment” throughout the manuscript to refer to the target dimension of the “current” investigation. Also, in the revised version of the manuscript, we better specify the target and the aim of the study.

3)    R2: The authors (line 34) define aesthetic experience as “the states of the mind and brain during the perception of works of art”. Does this mean that one cannot have an aesthetic experience to nature? Or to faces or bodies of other people? Does my appreciation of the beauty of nature or my partner not count as aesthetic? The authors conflate aesthetic experience with the appreciation of art. Please see Pearce et al. (2016) or other sources to clarify this issue.

Authors’ reply: The sentence refers to the operative definition in present study. Obviously, the experience of “Beauty” is not limited to the aesthetic experience of artworks. However, as it has been also reported in Pearce et al. 2016, artworks have been hypothesized as primary source of aesthetic experience (Anderson, 2000; Beardsley, 1983). We recognize that this sentence may be confusing so we have modified the first paragraph according to the Reviewer’s suggestion.

4)    R2: The authors (line 55) equate the “valence” of aesthetic experience with “like” or “dislike”. Not only does this seem arbitrary, it goes against much research showing that liking and disliking are related to the arousal dimension of affect (Berlyne, 1971; Marin & Leder, 2016).

Authors’ reply: here we are not referring to the affective valence of “arousing stimuli”, which may raise from watching affective pictures (as for example those of the International Affective Picture System, IAPS) or paintings (as used by Marin & Leder, 2016 who selected stimuli according to their arousal and complexity), neither we would “equate” the “valence” of aesthetic experience with “like” or “dislike”. We are referring to the “sign” of aesthetic judgment expressed by participants. In other words, here, we are referring to the degree of experienced attractiveness or ugliness. We have rephrased this sentence to avoid misunderstanding.

5)    R2:  The paper reports the effects of brain damage on the aesthetic appreciation of artworks, to use Pearce and colleagues’ (2016) terms. But the introduction overlooks much of the most significant work on the topic, especially the paper by Bromberger, Sternschein, Widick, Smith II and Chatterjee (2011). The authors do refer to two papers on the effects of neurodegenerative diseases, but they overlook others (e.g., Graham, Stockinger, & Leder, 2013; Halpern & O’Connor, 2013). It is also surprising, given the authors’ interest in the effects of lesions in left and right hemispheres, that they overlook the extensive literature on brain asymmetries in aesthetic experience and art appreciation (e.g., Beaumont, 1985; Coney & Bruce, 2004; Ellis & Miller, 1981; Levy, 1976; Zaidel, 1994, 2013, 2015; Zaidel & Kasher, 1989). A thorough coverage of the literature is good scientific practice, helps ground the study’s aims, and allows the proper assessment of its contribution.

Authors’ reply: We thank the Reviewer for suggesting us these useful references. We recognize the importance to include references to studies on neurodegenerative diseases (Graham, Stockinger, & Leder, 2013; Halpern & O’Connor, 2013) which have been added in the revised introduction section. Also, the discussion about the brain asymmetries in aesthetic experience has been updated with the suggested references (Zaidel, 1994, 2013, 2015; Zaidel & Kasher, 1989; Coney & Bruce, 2004). However, even if we retain the discussion of hand dominance interesting for aesthetic experience, we believe that it falls out of the main aim of the current work.

Methodological problems

6)    R2: The authors asked participants to rate the stimuli’s pleasantness. I would not say that this is an aesthetic assessment. It is an assessment of the valence dimension of the paintings. Despite the authors’ interchangeable use of this term with “aesthetic reaction”, “liking”, “aesthetic experience”, and so on, they are not synonymous. Affective valence is surely an important contributor to aesthetic experience, but it isn’t aesthetic experience. This is captured in prominent models of aesthetic experience (Chatterjee, 2003; Leder, Belke, Oeberst, & Augustin, 2004), where affective processes are components of a larger scheme. Thus, the conclusions from this paper refer not to aesthetic reactions or to visual aesthetic experience, but to affective evaluation of artworks. To present them otherwise is misleading.

Authors’ reply: We agree with the Reviewer about the need to be consistent in the terms used (see also the point above). We also agree that requiring participants to rate the pleasantness of the stimuli does not cover the entire range of processes which may be engaged in aesthetic experience. Also, we partially agree with the Reviewer that affective evaluation of artworks may be involved, but here we did not directly assess the “affective” dimension of aesthetic experience. Actually, we did not assess the positive or negative mood of the beholders or their emotional rating. However, as we requested participants to rate the pleasantness of artworks, we undoubtedly required participants to perform an Aesthetic Judgment task (see also the point 2 above). Thus, the present conclusions are certainly referring to this phenomenon. We now use this definition throughout the manuscript and restricted conclusion to it.

7)    a. R2: The authors provide the lesion reconstruction for only 9 of the 19 patients. I see this as the most damaging problems of all.

Authors’ reply: We agree that without providing the lesion reconstruction of the whole sample discussion of the neural underpinning of the aesthetic judgment is weak. However, our investigation is not merely devoted to the investigation of neural underpinnings of Aesthetic Judgment of ambiguous artworks. We aimed at assessing whether unilateral brain damage affects Aesthetic Judgment of ambiguous artworks and which are the “neuropsychological” underpinnings. In other words, we aimed at assessing whether visuo-perceptual, visuo-contructional and visuo-spatial deficits may affect Aesthetic Judgment of ambiguous artworks in unilaterally brain damaged patients. Furthermore, in this preliminary phase, with no previous studies assessing Aesthetic Judgment of ambiguous artworks, we retain not ethically correct to submit patients to an additional radiological exam. Anyway, we have provided extensive description of lesion location in Table 3, as it was reported in the medical record of each patient.

b. R2: In a nutshell, the authors cannot determine the precise brain region responsible for the effect they report, and therefore, they cannot provide the explanatory mechanism for it. The fact is that RBDP patients gave higher pleasantness ratings than control participants (they did not express higher aesthetic appreciation, as claimed by the authors in line 142). Why is this the case? Is this due to damage to brain regions involved in perception? Judgment? Affective processing? There is no way to tell because the location of the lesion is not appropriately described. The authors can only speculate, and do so in a most arbitrary way: “in our patients with right hemisphere damage appreciation of he local/parts in Arcimboldo’s paintings contributed to the aesthetic assessment”. Again, this is such a misleading statement that it borders on scientific fraud.

Authors’ reply: We would like to point out that, even if in presence of lesion reconstruction, any conclusions about the processes would be post hoc and misleading. Actually, it is not possible to draw conclusion about the underpinning cognitive processes starting from the neural data. The only way to do this is to have neuropsychological investigation of such abilities and processes. For this reason, we included a wide neuropsychological investigation of abilities we were interested in.

Concerning the discussion of the present results, in the revised version of the manuscript we have highlighted that the discussion of neural underpinnings is speculative and that further investigations are needed to neural bases of the observed effects.

c. R2: First, participants were assessing the valence of the artworks. Second, the authors themselves report no difference between RDBP and controls in the local or global levels of the Navon task. In the final paragraph of the conclusions, the authors overlook the fact that their main argument is based on speculation and present it as suggested by the results: “our results suggest that even in the absence of exploration or constructional deficits, lesions of the right hemisphere, somehow prevent the processing of the global/whole feature of the paintings …” There is nothing in the results that suggest this.

Authors’ reply: Starting from the Reviewer’s suggestions (the whole point 7) we have extensively revised the discussion section in order to avoid misunderstandings. Also, we discuss the absence of significant differences on the Navon Task, which are likely due to the use of accuracy rather than the reaction times as index of local/global processing. However, the use reaction times in the present study may be misleading due to the patients’ motor impairment.

8)    As a consequence of all the above, it is unclear what the scientific value of this study is. The authors themselves observe that “further studies are needed to obtain a clear localization of VAE, for example, by using voxel lesion symptom mapping”. This sentence is quite striking because this is precisely what Bromberger et al. (2011) published 6 years ago. In their study they had a precise mapping of right hemisphere lesions, which they were able to relate using voxel lesion symptom mapping to specific deficits in processing different kinds of attributes of the artworks. Science needs to be a cumulative enterprise. Ignoring previous work, such as Bromberg and colleagues’ (2011), can only lead us in circles. The authors should have designed their study based on the existing knowledge, trying to refine it, or even question it if they wish. Because the shortcomings of the paper under review here were solved 6 years ago, and the conclusions are based solely on speculation and disregarding part of the results, it seems to me that this work cannot make any meaningful contribution to our knowledge of the neural foundations of the appreciation of art.

Authors’ reply: when we stated that “further studies are needed to obtain a clear localization of VAE, for example, by using voxel lesion symptom mapping” we are referring to the Aesthetic Judgment of ambiguous portraits. Not to the VAE per se. We have rephrased this sentence to avoid misunderstandings.

Concerning the paper by Bromberger et al. (2011), we recognize that it is a valuable contribution to the neuropsychology of aesthetic experience, and we have included the reference to this work throughout the manuscript. However, these authors have not fully addressed all the possible questions of the cognitive neuroscience of art. Furthermore, no previous study has assessed the effect of the unilateral brain damage on Aesthetic Judgment of ambiguous portraits, nor the neuropsychological correlates of Aesthetic Judgment of ambiguous portraits. Thus, we believe that our contribution, even if weak from a “neuroradiological” point of view, assessed a different (even never-investigated issue) aspect of the cognitive neuroscience of art, that is the effect of artistic ambiguity and its “neuropsychological correlates”.

Minor issue:

R2: lines 81-83 please rephrase. This sentence is very confusing. It seems that the authors are suggesting that lesions in those brain regions lead to visual aesthetic experiences.

Authors’ reply: The reviewer is right in raising this issue. We have rephrased this sentence to avoid misunderstanding.

Round 2

Reviewer 2 Report

I believe the manuscript has been significantly improved and now warrants publication in Behavioral Sciences